# Reduction of Cystatin B results in increased cathepsin B activity in disomic but not Trisomy 21 human cellular and mouse models

Yixing Wu[1,2]*, Karen Cleverley[3], Frances K. Wiseman[1,2]*

**1** UK Dementia Research Institute at University College London, London, United Kingdom, **2** UCL Queen Square Institute of Neurology, Queen Square, London, United Kingdom, **3** Department of Neuromuscular Diseases, UCL Queen Square Institute of Neurology, Queen Square, London, United Kingdom

\* yixing.wu@ucl.ac.uk (YW); f.wiseman@ucl.ac.uk (FKW)

## Abstract

Down syndrome, resulting from trisomy of human chromosome 21, is a common form of chromosomal disorder that results in intellectual disability and altered risk of several medical conditions. Individuals with Down syndrome have a greatly increased risk of Alzheimer's disease (DSAD), due to the presence of the *APP* gene on chromosome 21 that encodes the amyloid-β precursor protein (APP). APP can be processed to generate amyloid-β, which accumulates in plaques in the brains of people who have Alzheimer's disease and is the upstream trigger of disease. Cathepsin B has potential roles in both APP processing and amyloid-β degradation and has been suggested to contribute to amyloid-β accumulation. An endogenous inhibitor of Cathepsin B, Cystatin B (*CSTB*), is encoded on chromosome 21. The abundance of this protein is increased in the brains of individuals with DSAD, which may be associated with a decrease in Cathepsin B activity compared to individuals who have Alzheimer's disease in the general population. Whether targeting *CSTB* can modulate Cathepsin B activity in the context of trisomy of chromosome 21 is unclear. Here we test if reducing CSTB can alter Cathepsin B activity in a mouse and a cellular model of trisomy of chromosome 21. We find that reducing CSTB abundance increases Cathepsin B activity in disomic controls but not in the presence of trisomy of chromosome 21. These findings offer new insights into the role of CSTB in regulating Cathepsin B activity.

## Introduction

Individuals with Down syndrome (DS) caused by trisomy of human chromosome 21 (Hsa21) experience cognitive impairment, craniofacial abnormalities and alterations in the function of their immune system and have a significantly increased risk of developing early onset Alzheimer's disease [1]. Alzheimer's disease (AD)-associated amyloid-β plaques develop in people with DS by age 40 [2]. This triggers a cascade of cellular and molecular changes in the brain that result in a greater increased risk of developing mid-life dementia [3]. By the age of 60, around two-thirds of the individuals with DS will have a clinical dementia diagnosis caused by AD (DSAD) [4]. Duplication of the Hsa21-located *APP* gene that encodes amyloid precursor

**Data Availability Statement:** The manuscript has been posted as a preprint on bioRxiv bioRxiv preprint doi: https://doi.org/10.1101/2024.04.10.

588836 All uncropped western blots are available at FigShare Wu et al 2024_raw_images.pdf. https://figshare.com/articles/figure/Wu_et_al_2024_raw_images_pdf/25567920.

**Funding:** Funding Statement F.K.W., is supported by the UK Dementia Research Institute (UKDRI-1014) through UK DRI Ltd, principally funded by the UK Medical Research Council. Y. W, is supported by an Alzheimer's Research UK Senior Research Fellowship (ARUK-SRF2018A-001 and ARUK-SRFEXT2022-001) awarded to F.K.W.

**Competing interests:** NO authors have competing interests.

protein (APP) causes an overproduction of amyloid-β, triggering its accumulation in the brain [5]. In addition, an extra copy of Hsa21 genes other than *APP*, also modulates the generation and accumulation of amyloid-β [6–11].

Cysteine protease, cathepsin B (CatB) [12], has important roles in protein degradation and cellular homeostasis [13, 14]. Increased abundance and activity of the enzyme is associated with AD in the general population [15–17], particularly with the accumulation of amyloid-β within the brain [18], and variation in the *CTSB* (CatB gene) is associated with the risk of late-onset AD [19]. CatB has been proposed to have a role in APP processing in the brain, including both the generation and catabolism of amyloid-β [17, 20, 21], but which mechanism predominates is unclear. Notably, decreased CatB activity is associated with dysfunction of lysosomes and accumulation of APP C-terminal fragments (CTFs) and amyloid-β [22, 23], phenotypes also associated with the early stages of DSAD [24]. We have previously found that CatB activity is reduced in the brain of individuals with DSAD compared with matched cases of EOAD from the general population [15]. Thus, normalising CatB activity may be a therapeutic strategy for the treatment of the early stages of DSAD.

The endogenous inhibitor of CatB, Cystatin B (CSTB), is encoded on Hsa21. Individuals who lack two functional copies of *CSTB*, develop a genetic form of epilepsy called Unverricht-Lundborg disease, that is progressive and associated with neurodegeneration. Cells isolated from individuals with Unverricht-Lundborg disease and animal models that lack functional *Cstb* exhibit reduced CatB activity [23, 25]. Trisomy of Hsa21 increases the abundance of CSTB in the brains of people with DSAD and in fibroblasts from individuals with DS [15]. Elevated levels of CSTB may contribute to the altered activity of CatB that is observed in DSAD, leading to dysregulated proteolysis and downstream effects on neuropathological features of AD. Although, an additional copy of *Cstb* is not sufficient to reduce CatB activity in the brain of DS mouse models, or in trisomy 21 fibroblasts under basal conditions [15, 26]. Whether lowering CSTB abundance in the context of trisomy of Hsa21 is sufficient to elevate CatB activity is unknown. Here we used human cellular and mouse models to investigate this.

## Materials and methods

### Mouse welfare and husbandry

Heterozygous *Cstb* knockout mice (*Cstb*$^{tm1b(EUCOMM)Wtsi}$ named here *Cstb*$^{+/-}$) (MGI MGI:5790639) were kindly supplied by the MRC Mary Lyon Centre, and maintained by mating male *Cstb*$^{+/-}$ to female C57BL/6J for one generation prior to crossing the progeny of this cross to Tc1 mice. Tc1 (Tc(HSA21)1TybEmcf/J) mice were taken from a colony maintained by mating Tc1 females (MGI: 3814712) to F1 (129S8 × C57BL/6) males. To generate the cohort studied here *Cstb*$^{+/-}$ males were mated with Tc1 females to produce four genotypes referred to as: wild-type (WT), *Cstb*$^{+/-}$, Tc1, and Tc1;*Cstb*$^{+/-}$. We ensured that no *Cstb*$^{-/-}$ animals were generated during our study because of adverse welfare outcomes associated with this genotype. In this study, all mice were housed in controlled conditions as per the Medical Research Council (MRC) and University College London (UCL)'s guidance. All experiments were conducted with approval from the Local Ethical Review panel and under License from the UK Home Office. Mice were semi-randomised by Mendelian inheritance of the genetically altered alleles into cages housing one sex, with at least two mice per cage. All mice were provided with bedding and wood chips, and continuous access to water. RM1 and RM3 chow were provided to breeding and stock mice, respectively by Special Diet Services, UK. Individually ventilated cages were in a specific-pathogen-free facility. Euthanasia of mice was carried out by exposing them to gradually increasing levels of $CO_2$ gas, and confirmation of death by dislocation of the neck, in compliance with the Animals (Scientific Procedures) Act issued in the United Kingdom in 1986.

## Genotyping

DNA was extracted from ear biopsies by the Hot Shot method [27]. Mice were genotyped using polymerase chain reaction (PCR) for the presence of human chromosome 21 (Tc1 specific primers f: 5′-GGTTTGAGGGAACACAAAGCTTAACTCCCA-3′ r: 5′-ACAGAGCTACAGCCT CTGACACTATGAACT-3′, control primers f: 5′-TTACGTCCATCGTGGACAGCAT-3′ r: 5′-TGGGCTGGGTGTTAGTCTTAT-3′) as described previously [28]. Mice were genotyped by PCR for the presence of the Cstb- (KO) and Cstb+ (WT) alleles with (Cstb-5arm-WT f: 5′-GTAGGGGGAGGTTCAGGGTA-3′, Cstb-Crit-WT r: 5′-GGCTGGCATGGAACTAAGCA-3′ and 5-KO r: 5′-GAACTTCGGAATAGGAACTTCG-3′).

## Cell culture

Cultured human fibroblasts derived from four individuals with Down syndrome (DS) (AG05397, AG07438, AG04823, and AG06922) and four euploid controls (GM05399, GM05565, GM05658 and GM05758) (Coriell Biorepository) were cultivated in Dulbecco's Modified Eagle Medium (DMEM). The DMEM was supplemented with 10% fetal bovine serum (FBS) and 100 units/ml of penicillin-streptomycin (Thermo Fisher Scientific). The cells were grown at 37˚C in an environment with 5% CO2.

When the cells reached about 70% confluency, they were harvested through trypsinization using Gibco™ Trypsin-EDTA (0.25%), phenol red at 37˚C. Subsequently, the cells were collected, pelleted, and washed three times using phosphate-buffered saline (PBS) before being homogenised.

## Dharmafect-mediated gene knockdown

DharmaconTM siRNA (Horizon) knockdown of CSTB in human fibroblasts was conducted as per the manufacturer's instructions with minor changes. In brief, a 5 μM siRNA solution (ON-TARGETplus Non-targeting Pool, Catalogue number: D-001810-10-05, ON-TARGET-plus GAPDH Control Pool (Human), Catalogue number: D-001830-10-05, or ON-TARGET-plus Human CSTB (1476) siRNA -SMARTpool, Catalogue number: L-017240-00-0005) was prepared using RNase-free water by diluting from the stock solution. Two separate tubes were used to dilute the siRNA (Tube 1) and the DharmaFECT (Horizon) transfection reagent (Tube 2) using serum-free medium. In Tube 1, a 200 μl diluted siRNA solution (for each well of a 6-well plate) was prepared in serum-free DMEM medium by combining 10 μl of 5 μM siRNA with 190 μl of serum-free medium. In Tube 2, a 200 μl diluted DharmaFECT transfection reagent solution was prepared in serum-free medium. The DharmaFECT reagent amounts used were 1 μl, 2.5 μl and 5 μl per well of a 6-well plate. The contents in Tube 1 and 2 were then mixed and incubated for 5 minutes at room temperature. The contents from Tube 1 were then added to Tube 2, and gently mixed by pipetting prior to a further incubation for 20 minutes at room temperature. After incubation, the transfection medium was added to each well of the 6 well plate (final concentration of siRNA: 25nM). The cells were incubated at 37˚C with 5% CO2 for 48 hours prior to analysis.

## Western blotting

To assess protein levels, the mouse cortex was homogenised using CB lysis buffer from the Cathepsin B Activity Assay Kit (Abcam, ab65300), with the addition of cOmplete™ Protease Inhibitor (Roche). Protein concentration was determined using a Bradford assay (Bio-Rad).

Mouse cortical homogenates were denatured using NuPAGE LDS Sample Buffer and NuPAGE™ Sample Reducing Agent (Thermo Fisher Scientific) at 95˚C for 5 minutes. They

were subsequently separated through SDS-polyacrylamide gel electrophoresis on a NuPAGE Novex 4–12% Bis-Tris gel (Thermo Fisher Scientific) at 150V for 50 minutes. The proteins were then transferred from the gel to a nitrocellulose membrane using the Trans-Blot Turbo™ Transfer System (Bio-Rad) at 25V, 2.5A for 7 minutes. Following transfer, the membranes were blocked using Intercept Blocking Buffer (LI-COR Bioscience) for 1 hour at room temperature.

For antibody probing, the membranes were incubated with primary antibodies overnight at 4˚C. The primary antibodies used were rabbit polyclonal anti-human Cystatin B (Abcam, ab236646, 1:2,000), rat monoclonal anti-mouse Cystatin B (Novus Biologicals, USA, #227818 (MAB1409), 1:2,000), rabbit polyclonal anti-cathepsin B (Abcam, ab92955, 1:1,000), rabbit monoclonal anti-GAPDH (Sigma, G9545, 1:5,000) and mouse monoclonal anti-β-actin antibody (Sigma-Aldrich, #A5441, 1:10,000). This was followed by incubation with secondary antibodies for 1 hour at room temperature. The secondary antibodies were IRDye 800CW Goat anti-Rabbit IgG (H + L) (1:10,000), Goat anti-Rat IRDye® 800CW IgG (H + L) and IRDye 680RD Goat anti-Mouse IgG (H + L) (1:10,000) (LI-COR Biosciences). Membranes were visualised using an Odyssey CLx Infrared Imaging System. The density of protein bands was quantified using ImageJ software.

For normalisation, the density of the CSTB or CatB protein band was divided by the density of the corresponding β-actin band run in the same lane. All uncropped western blots are available at FigShare Wu et al 2024_raw_images.pdf and as S1 Fig.

## Cathepsin B enzyme activity assay

The activity of CatB was examined in the cortex of 3-month-old mice or human fibroblasts using a Cathepsin B Activity Assay Kit (Abcam, #ab65300). The tissue or fibroblasts were homogenised in CB lysis buffer and then incubated on ice for 30 minutes before being centrifuged at 15,000 x g for 5 minutes at 4˚C. The resulting supernatant was transferred to a clean tube and protein concentration was determined using a Bradford assay (Bio-Rad). 200 µg tissue homogenate or 10 µg cell lysate was diluted in 50 µl CB lysis buffer and was used for the reaction with CatB Substrate (RR-amino-4-trifluoromethyl coumarin (AFC)).

To measure nonspecific cleavage, samples were treated with 50 µM inhibitors ALLM (Abcam, ab141446) or Z-Phe-Phe-FMK (Abcam, ab141386). The reaction mixture was then incubated at 37˚C in a microplate reader (Tecan), and the resulting fluorescent signal (excitation/emission = 400/505nm) was recorded every 90 seconds for 30 cycles by the microplate reader. The linear part of the reaction was determined, and the relative CatB activity in the sample was calculated by determining the average fluorescent output for each sample and subtracting the matched output from the inhibited reaction. Means of technical replicates were calculated for each individual sample, with biological replicate being used as the experimental unit. For CatB assays, 1 and 6 technical replicates were used for mouse cortex samples and human fibroblasts, respectively.

## Statistical analysis

All mouse experiments and data analyses were carried out blind to both genotype and sex. A unique 6-digit identifier was assigned to all mice and their homogenate tissue samples. Individual mouse or independent cell line were used as the experimental unit for all analysis.

The data are presented as group mean ± SEM, with individual datapoints for biological replicates. Data were analysed by ANOVA, using the mean of technical replicates, as indicated in the figure legends. For fibroblast studies variables of trisomy 21 status and treatment (control, *CSTB* knockdown or *GAPDH* knockdown) were used. For mouse studies variables of sex,

trisomy 21 status and $Cstb^{+/-}$ status were used. Pairwise comparisons for variables with more than two variants, with correction for multiple comparison as indicated in the figure legends, was undertaken when significant main effects or interactions were observed. Analyses were performed using GraphPad Prism 9 software (GraphPad Software) and SPSS version 26. Statistical significance was determined with a threshold of $p < 0.05$.

## Results

### CSTB knockdown increases CatB activity in disomic but not trisomy 21 human fibroblasts

To determine if targeting the endogenous inhibitor of CatB, CSTB, might be a viable strategy to increase cathepsin B activity in the context of trisomy of Hsa21, we used a siRNA approach to reduce CSTB abundance in human fibroblasts isolated from individuals with DS and matched euploid controls. We first optimized transfection conditions using a non-targeting negative control, GAPDH or CSTB siRNAs. A 1:400 dilution of transfection reagent was required to reduce GAPDH and CSTB abundance robustly in both disomic and trisomy 21 cells (Fig 1 and S2 Fig). To determine the effect of CSTB knockdown on CatB activity, a

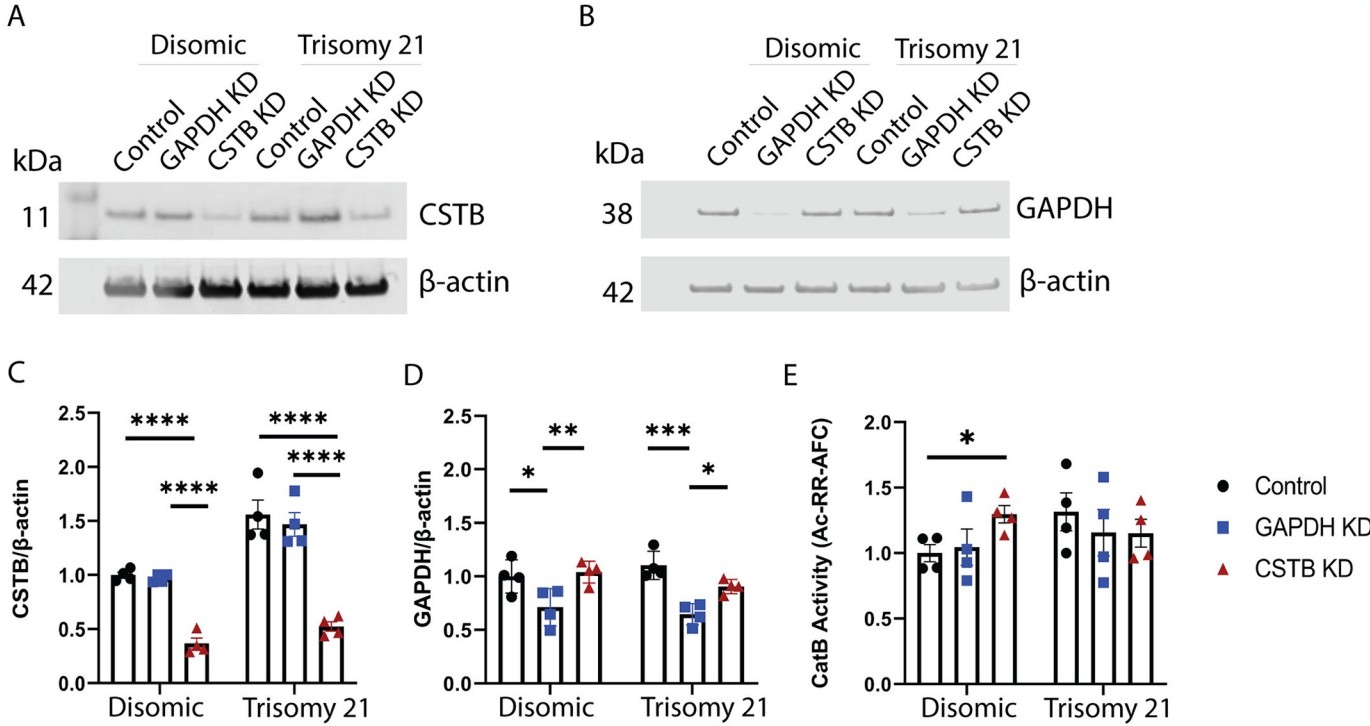

**Fig 1. *CSTB* knockdown and CatB activity in disomic and trisomy 21 human fibroblasts.** Western blot of CSTB (A) and GAPDH (B) normalised to β-actin in disomic and trisomy 21 human fibroblasts. (C) Trisomy 21 increased CSTB abundance in control and *GAPDH* KD groups (ANOVA main effect of Trisomy 21 F (1,18) = 41.92, p<0.0001, pairwise with Tukey correction control p<0.0001, *GAPDH* KD p = 0.0002). DharmaFECT-mediated knockdown reduced CSTB abundance compared with the control and the *GAPDH* KD groups (ANOVA main effect treatment F(2,18) = 73.50, p<0.0001, pairwise with Tukey correction control versus CSTB knockdown p<0.0001 (disomic and trisomy 21), GAPDH versus CSTB knockdown p<0.0001 (disomic and trisomy 21)). (D) DharmaFECT-mediated knockdown significantly reduced GAPDH compared with the control group and the *CSTB* KD group (ANOVA main effect treatment F(2,18) = 19.04, p<0.0001, pairwise with Tukey correction control versus GAPDH knockdown p = 0.0133 (disomic), p = 0.0002 (trisomy 21), GAPDH versus CSTB knockdown p = 0.0051 (disomic) p = 0.0259 (trisomy 21)). (E) *CSTB* knockdown significantly increases CatB activity compared with the control group and the *GAPDH* KD group in disomic, but not trisomic 21, fibroblasts as measured by the rate of cleavage of Ac-RR-AFC, corrected for nonspecific activity in samples inhibited by ALLM. (ANOVA interaction of treatment and trisomy F(2,12) = 4.621, p = 0.0325, pairwise with Tukey control versus CSTB knock-down p = 0.0418 (disomic)). Data are shown as ±SEM of group means for 4 disomic and 4 trisomy 21 lines (6 technical replicates for CSTB KD western blots, 4 technical replicates for GAPDH KD western blots and 2 technical replicates for CatB activity assay). *p<0.05, ***p<0.001, ****p<0.0001.

substrate (Ac-RR-AFC) cleavage assay was used and the mean rate of cleavage relative to the disomic control was calculated for all conditions. The siRNA-mediated reduction in CSTB abundance led to an increase in CatB enzyme activity in disomic, but not in trisomy 21 human fibroblasts (Fig 1E), despite the abundance of CSTB after knockdown not differing between disomic and trisomy 21 cells. Thus, targeting *CSTB* in trisomy 21 fibroblasts does not have a direct impact on CatB activity, in contrast to the effect of reducing this endogenous inhibitor on enzyme activity in disomic cells.

## CSTB knockdown does not affect the maturation of cathepsin B

Cathepsin B undergoes a maturation process by cleaving its proenzyme form to generate the mature active enzyme [29]. To test whether the reduction of CatB activity in disomic cells, mediated by *CSTB* knock-down, changed enzyme processing, the abundance of pro and mature CatB protein was measured by western blot (Fig 2A). Knocking down CSTB does not alter the protein level of pro-CatB, mature CatB or the mature CatB/pro-CatB ratio, compared with both untransfected controls and GAPDH knock-down in either disomic or trisomy 21 cells (Fig 2B–2D). These results suggest that the maturation of cathepsin B was unaffected by *CSTB* knockdown and thus changes to CatB activity in disomic fibroblasts occur via another process, likely mediated by a direct interaction between the endogenous inhibitor and enzyme.

## *Cstb* gene copy reduction leads to elevated CatB activity in the mouse brain but not in the presence of trisomy 21

To further understand the interaction of CSTB abundance, cathepsin B activity and trisomy of chromosome 21, we studied the effect of *Cstb* gene dose on CatB activity in a mouse model of DS. To do this we crossed the Tc1 mouse model of DS with $Cstb^{+/-}$ mice. The Tc1 mouse carries a copy of human chromosome 21, including an additional copy of human *CSTB*, alongside a normal complement of mouse chromosomes [28]. Thus, it contains 3 copies of the *CSTB/Cstb* gene and can be used to understand the effect of this on trisomy 21 biology. Importantly, this mouse model does not carry an additional functional copy of *APP* [30], and does not have raised abundance of APP in the brain [10], therefore it can be used to understand the effect of trisomy 21 independently of the effect of an additional copy of *APP*. The cross of Tc1 and $Cstb^{+/-}$ mice generated progeny with four genotypes: wildtype (WT) (2-copies of *Cstb*), Tc1 (3-copies of CSTB/*Cstb*), $Cstb^{+/-}$ (1-copy of *Cstb*) and Tc1;$Cstb^{+/-}$ ((2-copies of CSTB/*Cstb*).

We quantified the abundance of both mouse (Fig 3A) and human (Fig 3B) CSTB in total cortical proteins from these mice at 3-months of age. Human euploid and trisomic fibroblast homogenates were used to control for the specificity of the anti-mouse CSTB antibody (Fig 3A). Similarly, negligible signal was detected with the anti-human CSTB antibody in cortical samples from mice that did not express the human version of the protein (WT and $Cstb^{+/-}$) (Fig 3B). We found a significant decrease in mouse CSTB in $Cstb^{+/-}$ and Tc1;$Cstb^{+/-}$ cortices compared to WT and Tc1 controls (Fig 3A and 3D). Human CSTB levels were significantly higher in Tc1 and Tc1;$Cstb^{+/-}$ cortices than WT and $Cstb^{+/-}$ controls, with no difference observed between the Tc1 and Tc1;$Cstb^{+/-}$ samples (Fig 3E). Thus, reduction in *Cstb* gene copy number from two to one, or three to two, reduces the overall abundance of protein in the cortex of both the disomic and trisomy 21 mice respectively.

To investigate whether the reduction of *Cstb* has an effect on CatB maturation, the abundance of pro-CatB and mature CatB was quantified by western blot. No difference in pro-CatB or mature CatB protein abundance, or the mature cathepsin B/pro-CatB ratio was observed (Fig 3F–3H). Therefore, consistent with our findings in human fibroblasts, *Cstb* gene copy

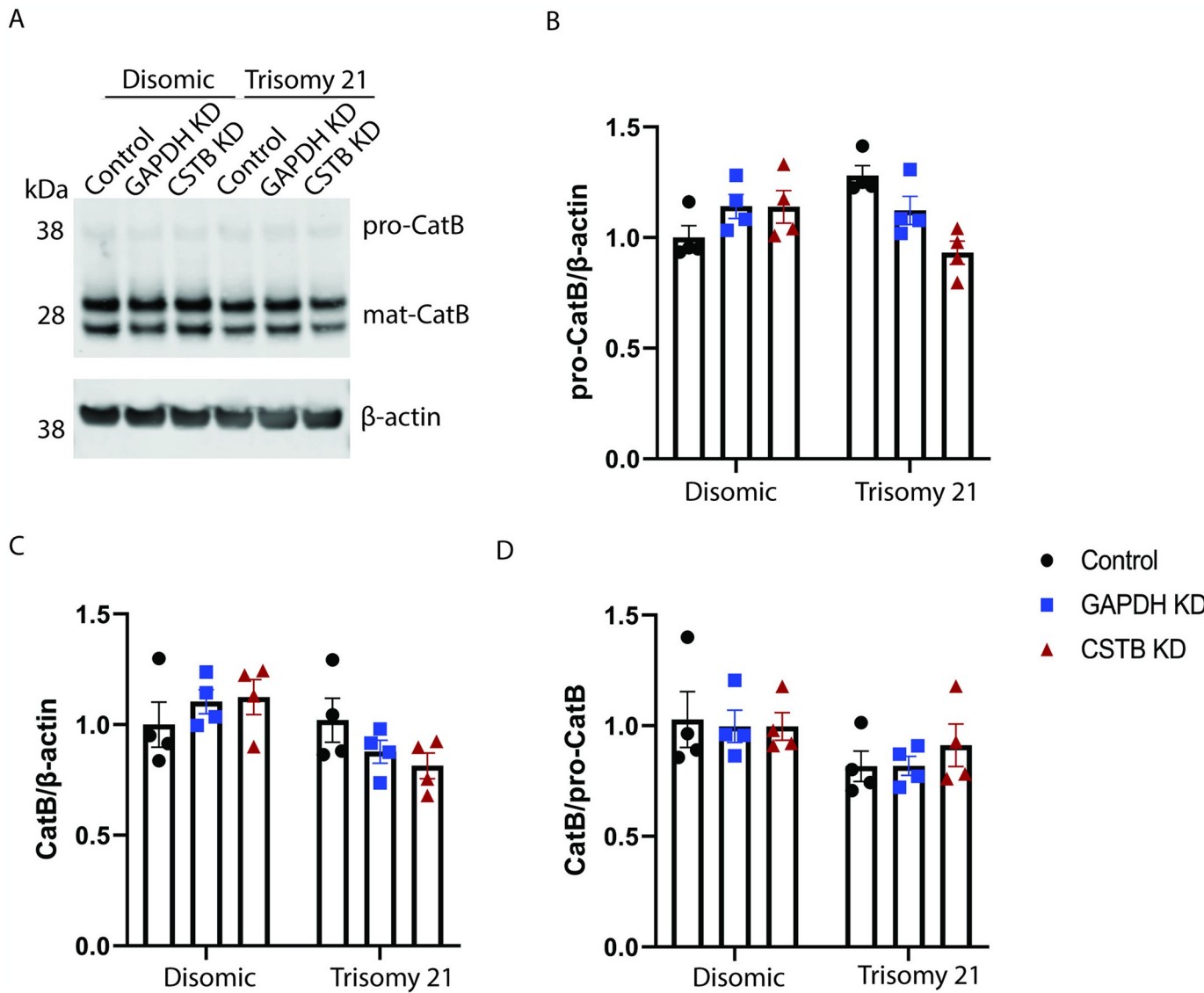

**Fig 2. Protein levels of pro and mature CatB in disomic and trisomy 21 human fibroblasts.** (A) Representative western blots of the proteins of pro-CatB and mature CatB, normalised to β-actin in disomic and trisomy 21 human fibroblasts. (B-D) DharmaFECT-mediated knockdown of *CSTB* does not alter the protein level of (B) pro-CatB, (C) mature CatB or (D) mature CatB/pro-CatB ratio compared with the control group and the *GAPDH* KD group. Data are shown ±SEM of group means for 4 disomic and 4 trisomy 21 lines (5 technical replicates for western blots). Data were analysed by two-way ANOVA followed by Tukey's post-hoc tests.

reduction does not alter CatB maturation in the cortex of either disomic or trisomy 21 mice at 3 months of age.

To determine how *Cstb* gene dose affected CatB activity, we undertook a biochemical cleavage assay on samples of mouse cortex at 3-months of age. The mean rate of CatB activity, corrected for non-specific activity using either ALLM or FMK inhibitors, was calculated for each genotype relative to the WT mean rate. CatB activity in $Cstb^{+/-}$ mice was significantly increased compared with WT, Tc1 and Tc1;$Cstb^{+/-}$ controls (Fig 3I and 3J), demonstrating that a reduction in the copy number of *Cstb* from two to one copies results in increased CatB activity within the disomic mouse brain. However, there was no difference in CatB activity between

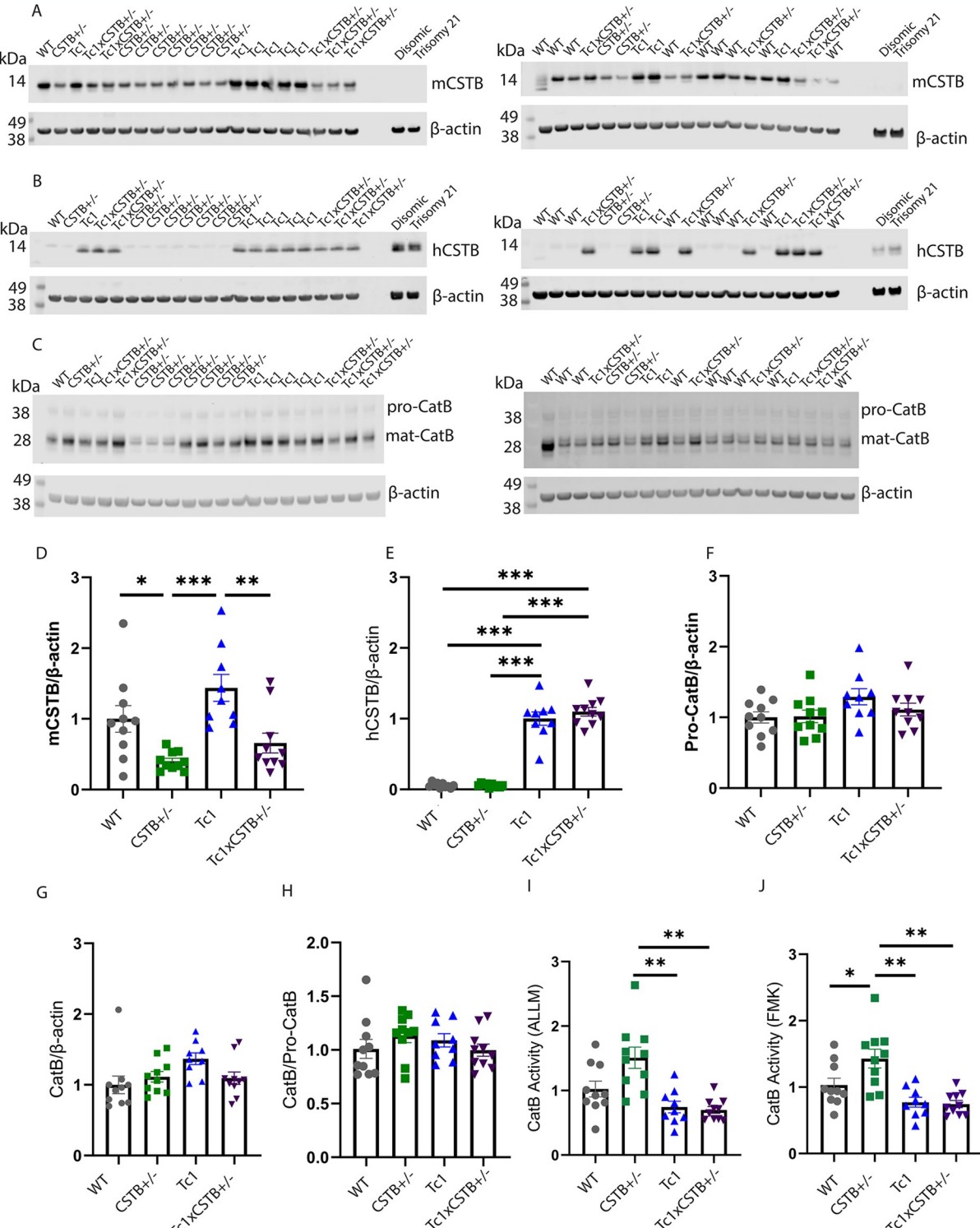

**Fig 3. CSTB and CatB abundance, and CatB activity in the brain of WT, *Cstb*^+/-, Tc1 and Tc1;*Cstb*^+/- cortex.** Representative western blots of (A) mouse CSTB (mCSTB), (B) human CSTB (hCSTB), (C) pro-CatB and mature CatB in cortex homogenate of WT, *Cstb*^+/-, Tc1 and Tc1; *Cstb*^+/- mice. The relative intensity of mCSTB, hCSTB, pro-CatB or mature CatB was quantified by normalising it with β-actin. (D) The abundance of mCSTB is lower in *Cstb*^+/- and Tc1;*Cstb*^+/- compared with WT and Tc1 controls (ANOVA *Cstb*^+/- genotype main-effect F(1,31) = 5.857, p = 0.022, pairwise comparison with Hochberg correction WT compared *Cstb*^+/- p = 0.043, Tc1 compared Tc1;*Cstb*^+/- p = 0.006, Tc1 compared *Cstb*^+/- p<0. 001). (E) The abundance of hCSTB was higher in Tc1 and Tc1;*Cstb*^+/- than WT and *Cstb*^+/- controls (ANOVA Tc1

genotype main-effect F(1,31) = 383.733, p<0.001, pairwise comparison with Hochberg correction WT/ *Cstb*$^{+/-}$ compared Tc1 p<0. 001, *Cstb*$^{+/-}$ compared Tc1;*Cstb*$^{+/-}$ p<0.001). No difference in the abundance of (F) proCatB, (G) mature CatB or (H) the mature CatB/pro-CatB ratio was detected in the WT, *Cstb*$^{+/-}$, Tc1 and Tc1; *Cstb*$^{+/-}$ cortex. (I) CatB activity as measured by biochemical assay (rate of cleavage of Ac-RR-AFC corrected for (I) ALLM or (J) FMK. (I) CatB activity differed between genotypes (ANOVA Tc1 genotype main-effect F(1,30) = 16.583, p<0.001, pairwise comparison with Hochberg correction *Cstb*$^{+/-}$ compared Tc1 p = 0.001, *Cstb*$^{+/-}$ compared Tc1;*Cstb*$^{+/-}$ p = 0.001). (J) CatB activity differed between genotypes (ANOVA Tc1 genotype main-effect F(1,30) = 16.106, p<0.001, pairwise comparison with Hochberg correction WT compared *Cstb*+/- p = 0.048, *Cstb*+/- compared Tc1 p = 0.001, *Cstb*+/- compared Tc1;*Cstb*+/- p = 0.001). (A-H) 10 WT (5 female 5 male), 10 *Cstb*$^{+/-}$ (3 female, 7 male), 9 Tc1 (3 female, 6 male) and 10 Tc1;*Cstb*$^{+/-}$ (4 female, 6 male) mice. (I, J) 10 WT (5 female, 5 male), 10 *Cstb*$^{+/-}$ (3 female, 7 male), 9 Tc1 (3 female, 6 male) and 9 Tc1;*Cstb*$^{+/-}$ (4 female, 5 male) mice. 2–3 technical replicates for western blots and 2 technical replicates for CatB activity assay). *p<0.05, **p<0.01, ***p<0.001, error bars SEM.

the Tc1 and Tc1;*Cstb*$^{+/-}$ groups (Fig 3I and 3J). Thus, reducing the *Cstb* gene copy number from three to two copies in the presence of trisomy 21 in the young adult cortex does not modify CatB activity.

## Discussion

The interaction between CSTB and CatB plays an important role in balancing proteolytic activity within cells [25]. In this study, we investigated whether altering this balance by either reducing protein levels of CSTB using siRNA-mediated knockdown or reducing the number of copies of the *Cstb* gene, can lead to an increase in CatB activity, or a change in CatB maturation, in the context of trisomy of Hsa21. Our results showed that in disomic human fibroblasts, knocking down *CSTB* increases CatB activity. Similarly, reducing *Cstb* from two to one copy in mice also leads to an increase in CatB activity in the young adult cortex, consistent with a previous report [23]. In contrast, in the presence of trisomy 21, knocking down CSTB in human fibroblasts or lowering *Cstb* gene dose in a mouse model of DS, from three to two copies, does not alter CatB activity.

Previously, we have shown that an additional copy of *Cstb/CSTB* is not sufficient to alter CatB activity in a range of DS preclinical models [15, 26]. Here, we show that in the context of trisomy of Hsa21, less CSTB is also not sufficient to modify enzyme activity, in contrast to the effect of the copy number of this gene in disomic cells and brain. This may be the result of other regulators of CatB activity being differentially regulated by trisomy 21. For example, cystatin C (CST3) also modulates CatB activity [21, 31, 32], and has been reported to be upregulated by trisomy 21 [33]. However, in the Ts2Cje mouse model of DS overexpressing CST3 improved endosomal morphology and alleviated behavioural defects but does not alter CatB activity [34], indicating that increased abundance of this cystatin may also be insufficient to change CatB activity in the context of DS.

We have previously shown that in human temporal cortex CatB activity is lower in cases of DSAD compared with cases of early-onset AD from the general population [15]. The difference between these data and the findings in our preclinical models indicates that the effect of trisomy 21 on CatB activity could depend on the development of AD neuropathology, which includes accumulation of misfolded amyloid-β and tau. Further research in DSAD model systems is warranted to understand this complex biology. However, here we highlight that in trisomy 21 preclinical systems, in the absence of features of AD neuropathology, targeting CSTB is not sufficient to increase CatB activity. These data indicate that trisomy 21 alters the effect of lowering CSTB abundance independently of AD neuropathology.

In addition to cystatins, CatB activity is also regulated by the processing of the enzyme. Here we show, consistent with our previous work, that CatB processing as measured by the pro/mature ratio is not affected by trisomy 21 [15]. Thus, another mechanism likely results in the insensitivity of CatB activity to CSTB abundance in the context of trisomy 21. Trisomy 21

results in perturbations to endo-lysosomal biology, in part because of an effect of APP-CTF on v-ATPase acidification [35]. Notably, the Tc1 mouse model does not have an additional functional copy of *APP*, thus raised APP-CTF is unlikely to be the cause of the insensitivity to *Cstb* gene dose that we observed.

In our DS mouse model we reduced the *CSTB/Cstb* gene from three to two copies, rather than three copies to one copy. This is a limitation of our study, and reduction to only one copy of the gene, by deletion of both copies of mouse *Cstb*, may be required to modify CatB activity in the brain. We note that the deletion of both copies of *Cstb* in the mouse, recapitulates many features of Unverricht-Lundborg disease, including seizures, progressive ataxia and neurodegeneration [36, 37]. Thus, in our preclinical systems we aimed to reduce but not completely eliminate CSTB protein, to avoid inducing features of Unverricht-Lundborg disease.

In summary, results from our study indicate the complexity of the relationship between CSTB, CatB, and trisomy 21. While reducing the abundance of CSTB increased CatB activity in disomic human fibroblasts and mouse brain, this effect is not replicated in the presence of trisomy 21, suggesting that *CSTB* gene dose does not contribute to the regulation of CatB activity in people who have DS. Thus, targeting CSTB is unlikely to be a useful strategy to normalize CatB activity in the context of DS.

## Supporting information

**S1 Fig. Uncropped western blot files for all figures in this manuscript.**
(PDF)

**S2 Fig. *CSTB* and *GAPDH* knockdown in disomic and trisomy 21 human fibroblasts using different concentrations of DharmaFECT transfection reagent.** (A-B) Western blot of CSTB and GAPDH normalised to β-actin in disomic and trisomy 21 human fibroblasts in (A) 1:2,000 dilution of DharmaFECT reagent or (B) 1:800 dilution DharmaFECT reagent. (C) 1:2,000 dilution of DharmaFECT-mediated *CSTB* knockdown reduced CSTB abundance compared with the control group (p = 0.0058) in the disomic, but not in the trisomy 21 fibroblasts. (D) 1:800 dilution of DharmaFECT-mediated *CSTB* knockdown reduced CSTB abundance in the *CSTB* KD group compared with the *GAPDH* KD group (p = 0.0139), but not compared to the control group in the trisomic 21 human fibroblasts or the disomic group. (E) 1:2000 dilution of DharmaFECT-mediated *GAPDH* knockdown does not affect GAPDH abundance in either disomic or trisomy 21 fibroblasts. (F) 1:800 dilution of DharmaFECT-mediated *GAPDH* knockdown reduced GAPDH abundance in disomic fibroblasts compared to control (p = 0.002) and CSTB (p = 0.0028) groups. In the trisomy 21 fibroblasts, *GAPDH* knockdown reduced GAPDH abundance compared with the control (p = 0.0025), but not the *CSTB* knockdown group. Data are shown as ±SEM of group means for 4 disomic and 4 trisomy 21 lines (1 technical replicate for western blots). Data were analysed by two-way ANOVA followed by Tukey's post-hoc tests, *p<0.05, ***p<0.001, ****p<0.0001.
(TIF)

## Acknowledgments

We thank Dr. T. Cunningham (MRC Mouse Genetics Unit and Institute of Prion diseases, University College London) for help with this project. For the purpose of Open Access, the author has applied a CC-BY public copyright licence to any Author Accepted Manuscript version arising from this submission.

## Author Contributions

**Conceptualization:** Frances K. Wiseman.

**Formal analysis:** Yixing Wu, Frances K. Wiseman.

**Funding acquisition:** Frances K. Wiseman.

**Investigation:** Yixing Wu, Frances K. Wiseman.

**Methodology:** Yixing Wu, Karen Cleverley, Frances K. Wiseman.

**Project administration:** Frances K. Wiseman.

**Resources:** Frances K. Wiseman.

**Software:** Frances K. Wiseman.

**Supervision:** Frances K. Wiseman.

**Validation:** Yixing Wu.

**Writing – original draft:** Yixing Wu, Frances K. Wiseman.

**Writing – review & editing:** Yixing Wu, Karen Cleverley, Frances K. Wiseman.

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
