## [Decision Letter · Decision Letter 0]

9 Aug 2024

PONE-D-24-15390Reduction of Cystatin B results in increased cathepsin B activity in disomic but not Trisomy21 human cellular and mouse modelsPLOS ONE

Dear Dr. Wu,

Thank you for submitting your manuscript to PLOS ONE. After careful consideration, we feel that it has merit but does not fully meet PLOS ONE’s publication criteria as it currently stands. Therefore, we invite you to submit a revised version of the manuscript that addresses the points raised during the review process.

**ACADEMIC EDITOR:**The reviewer raised several issues that need to be addressed. Please include a point-by-point response to reviewer's comments. Please note that the manuscript will be judged based on PLOS ONE's publication criteria and not, for example, on novelty or perceived impact.

We look forward to receiving your revised manuscript.

Kind regards,

Giuseppe Barisano, M.D., Ph.D.

Guest Editor

PLOS ONE

Journal Requirements:

"Funding Statement

F.K.W., is supported by the UK Dementia Research Institute (UKDRI-1014) through UK DRI Ltd, principally funded by the UK Medical Research Council. Y. W, is supported by an Alzheimer’s Research UK Senior Research Fellowship (ARUK-SRF2018A-001 and ARUK-SRFEXT2022-001) awarded to F.K.W."

"NO authors have competing interests"

6. We notice that your supplementary [S1 Fig.] are included in the manuscript file. Please remove them and upload them with the file type 'Supporting Information'. Please ensure that each Supporting Information file has a legend listed in the manuscript after the references list.

Reviewers' comments:

Reviewer's Responses to Questions

**Comments to the Author**

1. Is the manuscript technically sound, and do the data support the conclusions?

Reviewer #1: Partly

2. Has the statistical analysis been performed appropriately and rigorously? 

Reviewer #1: Yes

3. Have the authors made all data underlying the findings in their manuscript fully available?

Reviewer #1: Yes

4. Is the manuscript presented in an intelligible fashion and written in standard English?

Reviewer #1: Yes

5. Review Comments to the Author

Reviewer #1: The manuscript by Wu et al. submitted to PlosONE focuses on the effect of Cystatin-B (CSTB) downregulation on Cathepsin B (CatB) activity and abundance. In vitro, in disomic human fibroblasts, CSTB knock-down enhances CatB activity (Fig. 1), but not protein levels (Fig. 2), while no effect was found in Down syndrome (DS) cells (Fig. 1-2). Similarly, lowering the levels of CSTB in vivo enhances CatB activity in the cerebral cortex of WT (but not trisomic) mice (Fig. 3).

The manuscript is well written, and the statistical analyses are rigorous. However, some concerns in the interpretation of results, novelty, and overall rationale of the study dampen the enthusiasm.

1. Interpretation of data for Figures 3I and 3J are exaggerated. Tc1 mice have three copies of the Cstb gene, whereas Tc1xCstb+/-, as well as WT mice, have two copies. Therefore, only Cstb+/- mice can be considered a real downregulation model of Cstb in vivo, given that they bear less functional Cstb genes (only one) than wild-type conditions. With this said, it is unsurprising that CatB activity is the same in Tc1xCstb+/- and WT mice. Similarly, the lack of a difference between Tc1 and Tc1xCstb+/- mice can be explained simply by an insufficient reduction in CSTB levels in Tc1xCstb+/- mice, which still carry the same amount of CSTB than WT mice. For these reasons, in my opinion these experiments are somewhat inconclusive, and should not be considered the equivalent in vivo of what shown in cells in vitro - in Figure 1, the downregulation of Cstb is similar between WT and DS cells, while this is not the case in Cstb+/- and Tc1xCstb+/- murine brains. I suggest repeating the experiments shown in Figure 3 in Tc1xCstb-/- (Cstb-KO) mice, in which only one functional gene is found.

2. The evidence that reduced CSTB levels correlate with higher CatB activity in wild-type, basal conditions is not novel, and described before (e.g., Rinne et al., PMID: 12452481). The evidence that no correlation exists between CSTB levels and CatB in the context of DS was also demonstrated before by the same authors of this manuscript (e.g., Wu et al., PMID: 12452481). Therefore, I do not understand where the novelty of the findings reported in this paper is. What do these new experiments tell us that was not known previously? This should be fully discussed.

3. In previous research from the same group, as quickly mentioned at point 2, no differences were found in CatB activity when comparing DS with WT cells (Wu et al., PMID: 12452481). Therefore, the rationale for the current study remains a little obscure. Why did the authors compare the effects of Cstb knockdown on CatB activity in DS and WT cells, if they already knew that there is no effect of the same gene on CatB in DS? The same authors reported instead a difference between DSAD vs AD without trisomy (Wu et al., PMID: 12452481). Accordingly, one would expect a comparison in CatB activity when Cstb is downregulated in DSAD vs AD, not between DS vs diploid, but this is not shown. This point is crucial for the overall project and should be thoroughly explained.

6. PLOS authors have the option to publish the peer review history of their article (what does this mean?). If published, this will include your full peer review and any attached files.

Reviewer #1: No

---

## [Author Response · Author response to Decision Letter 0]

27 Nov 2024

Dear PLoS One Editorial Board,

We are submitting our revised research article entitled “Reduction of Cystatin B results in increased cathepsin B activity in disomic but not Trisomy21 human cellular and mouse models” for consideration by PLoS One. In this study, we used human fibroblasts and mouse models to determine if targeting CSTB/cystatin B in the context of trisomy of human chromosome 21 (Hsa21) is sufficient to elevate Cathepsin B activity. 

Editorial Requested Changes

We can confirm that the manuscript now meets PLoS style requirements, including the order of the sections and we have inserted the Figure legends directly after the paragraph in which they are first referred as directed. 

We have added the statement “The funders had no role in study design, data collection and analysis, decision to publish, or preparation of the manuscript" as requested to the funding statement lines 449-451.

We have completed the competing interests form as requested.

We have added all uncropped western blots for the manuscript as an additional supplementary figure (S1 Fig), as well as the previous Figshare upload of these data. 

We have deleted S1 Fig. from the manuscript text and uploaded it as a Supporting Information File type.

Response to Reviewer’s comments

We thank the reviewer for the constructive and helpful suggestions, please find below a response to each of the points raised.

• Reviewer 1 point 1: Tc1 mice have three copies of the Cstb gene, whereas Tc1xCstb+/-, as well as WT mice, have two copies. Therefore, only Cstb+/- mice can be considered a real downregulation model of Cstb in vivo, given that they bear less functional Cstb genes (only one) than wild-type conditions….. . I suggest repeating the experiments shown in Figure 3 in Tc1xCstb-/- (Cstb-KO) mice, in which only one functional gene is found.

We have edited both the results (lines 367, 370) and discussion (lines 380, 385 and 419-426) sections to highlight this limitation of our in vivo studies. Unfortunately, we are not permitted by our UK government animal licence to undertake the suggested experiment. Cstb-KO mice develop myoclonus, ataxia and neurodegeneration (lines 422-424), and we are not permitted to generate mice without a functional copy of Cstb under the terms of our animal research licence because of the high welfare burden these animals experience.

• Reviewer 1 point 2. The evidence that reduced CSTB levels correlate with higher CatB activity in wild-type, basal conditions is not novel, and described before (e.g., Rinne et al., PMID: 12452481). The evidence that no correlation exists between CSTB levels and CatB in the context of DS was also demonstrated before by the same authors of this manuscript (e.g., Wu et al., PMID: 12452481). Therefore, I do not understand where the novelty of the findings reported in this paper is. What do these new experiments tell us that was not known previously? This should be fully discussed.

As the reviewer highlights, the effect on CatB activity of homozygous loss of function of CSTB or Cstb genes has been previously shown in disomic preclinical models, including patient cell lines and mouse models. We have added this important information to the introduction along with the suggested reference (lines 67-70). As stated in our manuscript, our principal novel finding is that reducing CSTB abundance in the context of trisomy of Hsa21 does not reduce CatB activity. Please see lines 388-392 of the discussion regarding this “Previously, we have shown that an additional copy of Cstb/CSTB is not sufficient to alter CatB activity in a range of DS preclinical models [15, 25]. Here, we show that in the context of trisomy of Hsa21, less CSTB is also not sufficient to modify enzyme activity, in contrast to the effect of the copy number of this gene in disomic cells and brain.” 

• Reviewer 1 point 3. In previous research from the same group, as quickly mentioned at point 2, no differences were found in CatB activity when comparing DS with WT cells (Wu et al., PMID: 12452481). Therefore, the rationale for the current study remains a little obscure. Why did the authors compare the effects of Cstb knockdown on CatB activity in DS and WT cells, if they already knew that there is no effect of the same gene on CatB in DS? The same authors reported instead a difference between DSAD vs AD without trisomy (Wu et al., PMID: 12452481). Accordingly, one would expect a comparison in CatB activity when Cstb is downregulated in DSAD vs AD, not between DS vs diploid, but this is not shown. This point is crucial for the overall project and should be thoroughly explained.

Our previous work in human post-mortem tissues demonstrated that CatB activity was reduced in DSAD compared with AD in the general population (Wu et al., PMID: 12452481), please see lines 61-63 of the introduction. However, the cause and consequence of this on DSAD development is currently unclear. Thus we aimed to determine if we could modify CatB activity in a trisomy 21 context. We note that the modelling of DSAD in preclinical systems is challenging, and currently only aspects of DSAD can be effectively modelled in animal and human cellular models. Thus prior to investigating if targeting CSTB could elevate CatB activity in DSAD model systems, we first undertook the proof-of-prinicipal experiments in a simpler model system. We report these studies here, in which we aimed to determine if targeting CSTB/Cstb in a trisomy 21 context would increase enzyme activity. To our surprise, we found that in contrast to disomic cells that when CSTB abundance is reduced in trisomy 21 model systems we observed no effect on CatB activity. Collectively, these data indicate that trisomy 21 likely alters CatB activity independently of CSTB and that an alternative approach will be required to modulate enzyme activity in DSAD. We have added a discussion of these issues to the manuscript (lines 400-409).

Yours sincerely,

Frances K. Wiseman PhD.

---

## [Decision Letter · Decision Letter 1]

17 Dec 2024

Reduction of Cystatin B results in increased cathepsin B activity in disomic but not Trisomy21 human cellular and mouse models

PONE-D-24-15390R1

Dear Dr. Wu,

We’re pleased to inform you that your manuscript has been judged scientifically suitable for publication and will be formally accepted for publication once it meets all outstanding technical requirements.

Kind regards,

Stephan N. Witt, Ph.D.

Academic Editor

PLOS ONE

Additional Editor Comments (optional):

Reviewers' comments:

Reviewer's Responses to Questions

**Comments to the Author**

1. If the authors have adequately addressed your comments raised in a previous round of review and you feel that this manuscript is now acceptable for publication, you may indicate that here to bypass the “Comments to the Author” section, enter your conflict of interest statement in the “Confidential to Editor” section, and submit your "Accept" recommendation.

Reviewer #1: All comments have been addressed

2. Is the manuscript technically sound, and do the data support the conclusions?

Reviewer #1: Yes

3. Has the statistical analysis been performed appropriately and rigorously? 

Reviewer #1: Yes

4. Have the authors made all data underlying the findings in their manuscript fully available?

Reviewer #1: Yes

5. Is the manuscript presented in an intelligible fashion and written in standard English?

Reviewer #1: Yes

6. Review Comments to the Author

Reviewer #1: (No Response)

7. PLOS authors have the option to publish the peer review history of their article (what does this mean?). If published, this will include your full peer review and any attached files.

Reviewer #1: No

---

## [Editor Report · Acceptance letter]

10 Jan 2025

PONE-D-24-15390R1 

PLOS ONE

Dear Dr. Wu, 

I'm pleased to inform you that your manuscript has been deemed suitable for publication in PLOS ONE. Congratulations! Your manuscript is now being handed over to our production team.

Kind regards, 

on behalf of

Dr. Stephan N. Witt 

Academic Editor

PLOS ONE